# Detecting Opioid Use Disorder in Health Claims Data with Positive Unlabeled Learning

[1]Praveen Kumar, [1]Fariha Moomtaheen, [1]Scott A. Malec, [1]Jeremy J. Yang, [1]Cristian G. Bologa,
[1]Kristan A Schneider, [1]Yiliang Zhu, [2]Mauricio Tohen, [2]Gerardo Villarreal, [1]Douglas J. Perkins,
[3]Elliot M. Fielstein, [3]Sharon E. Davis, [3]Michael E. Matheny, [1]Christophe G. Lambert

[1]Department of Internal Medicine, University of New Mexico, Albuquerque, USA
[2]Department of Psychiatry and Behavioral Sciences, University of New Mexico, Albuquerque, USA
[3]Department of Biomedical Informatics, Vanderbilt University Medical Center, Nashville, USA

*Abstract*—Accurate detection and prevalence estimation of behavioral health conditions, such as opioid use disorder (OUD), are crucial for identifying at-risk individuals, determining treatment needs, monitoring prevention and intervention efforts, and recruiting treatment-naive participants for clinical trials. The availability of extensive health data, combined with advancements in machine learning (ML) frameworks, has enabled researchers to employ various ML techniques to predict or identify OUD within patient health data. Ideally, we could directly estimate the prevalence, or the proportion of a population with a condition over time. However, underdiagnosis and undercoding of conditions in patient health records make it challenging to determine the true prevalence of these conditions and to identify at-risk patients with less severe conditions who are more likely to be missed. Consequently, patients without diagnoses may comprise positive and negative examples for a given condition. Treating all undiagnosed (uncoded) patients as negative when applying ML methods can introduce bias into models, affecting their predictive power. To address this issue, we employed Positive Unlabeled Learning Selected Not At Random (PULSNAR), a Positive and Unlabeled (PU) learning technique, to estimate the probability of a given patient having OUD during a time window and the overall population prevalence of OUD. In a sample of 3,342,044 commercially insured US patients with at least one opioid prescription filled, PULSNAR estimated that 5.08% of patients have a cumulative prevalence of OUD over a 2-5 year observation period, compared to the 1.35% with a recorded OUD diagnosis, with 73.5% of cases not diagnosed/coded. The prevalence estimates provided by PULSNAR are consistent with those reported in other studies.

*Index Terms*—Machine learning, opioid use disorder, OUD, Positive Unlabeled learning, PU learning, PULSCAR, PULSNAR, SCAR, SNAR

## I. INTRODUCTION

### A. Background and significance

Opioid Use Disorder (OUD) is a chronic mental health condition characterized by prolonged use of opioids despite adverse consequences, leading to significant clinical distress or impairment [1], [2]. This disorder includes a spectrum of opioids, ranging from synthetic ones like fentanyl to prescription pain relievers such as oxycodone, hydrocodone, codeine, and morphine, to the illegal drug heroin, among others [3]. OUD manifests through a strong urge to use opioids, increased

This research was supported by the US National Institutes of Health grants R01MH129764 and R00LM013367, and infrastructure support from the National Center for Advancing Translational Sciences grant UL1TR001449.

tolerance to their effects, and withdrawal symptoms upon discontinuation [1]. Over time, individuals develop a two-fold dependence on these substances, physical and psychological, which results in withdrawal symptoms if they abruptly stop opioid intake [2], [4]. The untreated progression of OUD can result in significant alterations to brain structure and function [5], leading to profound social, economic, and health repercussions [2].

The opioid crisis continues to be a significant global public health issue [6]. Opioids were responsible for nearly 80% of the approximately 600,000 substance-related deaths worldwide in 2019, with around 125,000 deaths resulting from opioid overdoses [7]. In the US, 107,941 drug overdose deaths occurred in 2022, with opioids contributing to 81,806 (75.8%) of these deaths [8]. The economic burden of OUD and fatal opioid overdoses in the US was estimated at $1.02 trillion in 2017 [9], rising to nearly $1.5 trillion in 2020 [10].

The ripple effects of OUD extend to individuals, their families, and the broader community, causing a substantial increase in morbidity and mortality rates [1], [11]. Earlier detection is crucial for identifying at-risk individuals and addressing their treatment needs, which is essential for mitigating OUD's social and economic consequences [1]. By detecting the disorder early, morbidity and mortality associated with OUD can be reduced [11], [12]. However, the prevalent underdiagnosis and undercoding of this disorder and its co-occurring conditions in electronic health records (EHRs) and claims data [13]–[18] can impede public health initiatives, screening efforts, and the identification of health disparities, resulting in missed opportunities to mitigate the harms of these conditions.

Applying traditional positive and negative binary classification techniques to OUD data can result in a biased model [19], as uncoded examples may contain positive instances of OUD due to a lack of diagnosis and undercoding. This makes it difficult to fully characterize the problem, estimate prevalence, understand disparities, and accurately predict at-risk individuals. Positive and Unlabeled (PU) learning presents a more appropriate approach for handling such data, as it accounts for the presence of positive instances within the unlabeled set.

Our study aims to leverage PU learning to estimate both the probability of individual OUD and the overall prevalence

among individuals who have been exposed to opioids. Additionally, we analyze the differences in coded and imputed OUD diagnoses across different US states, age groups, and sexes. To the best of our knowledge, this is the first study to employ PU learning on opioid-related data to estimate the prevalence or predict OUD. By addressing the inherent underdiagnosis and undercoding of OUD through the PU learning method, our approach has the potential to provide more accurate estimates of individual risk and overall prevalence, enabling better characterization of the problem. Furthermore, analyzing demographic and geographic variations in OUD diagnoses can identify potential disparities for intervention.

### B. Related work

The surge in healthcare data and advanced machine learning (ML) frameworks has enabled the development of ML models to address various aspects of the opioid crisis [4], [12], [20]–[34]. Researchers have developed these models using EHR and claims data [4], [12], [20]–[34]; some studies focused on structured data [4], [12], [20]–[29], while others analyzed unstructured data [30]–[34] to predict or detect different dimensions of the crisis.

A number of prior publications have evaluated AI/ML methods on EHR and claims data for OUD prediction, including Logistic Regression (LR) [22], [24]–[26], Gradient Boosting Trees algorithm (XGBoost) [20], [26], Decision Tree (DT) [22], [24], Elastic Net (EN) [21], Random Forest (RF) [4], [21]–[23], [25], Gradient Boosting Machine (GBM) [21], [22], [25], and Cox Regression [29], as well as multiple types of neural networks, including Artificial Neural Networks (ANN) [25], [27], Deep Neural Network (DNN) [21], [28], and Long Short-Term Memory (LSTM) [12] models. These studies have employed ML approaches to predict OUD across diverse populations, such as Medicare beneficiaries, individuals who have undergone hip arthroscopy, US adolescents, and others. While these methods demonstrated that ML models could predict OUD with good accuracy, their generalizability is limited due to their training on specific population subsets, potentially not representing broader demographics. Several of these studies have acknowledged the potential impact of class imbalance on the model's classification performance. However, previous studies did not primarily focus on strategies to address this issue, except for one investigation [22], which utilized the Synthetic Minority Oversampling Technique (SMOTE) [35] to oversample the minority class. Despite recognizing the underdiagnosis of OUD in health data, these methods treated all undiagnosed cases as negative examples in the model without explaining how to address this potential source of bias.

There has also been prior work evaluating the utility of incorporating narrative text for the prediction and identification of OUD, with many of the AI/ML methods being used on narrative text as the sole data source or in combination with more structured data [30]–[34]. While these methods demonstrated the potential of leveraging free text-based NLP to enhance the identification and prediction of OUD, their development was constrained by key factors. These included

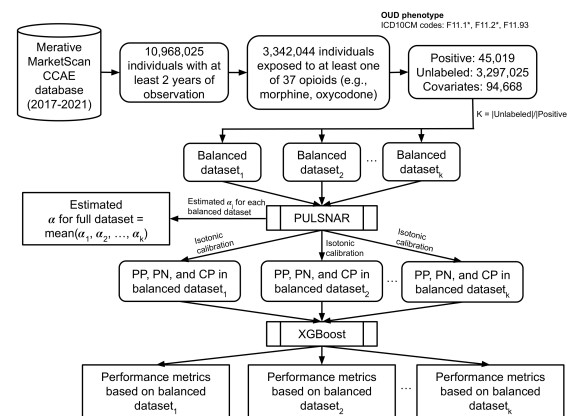

Fig. 1. **Steps to estimate the proportion of uncoded OUD using PULSNAR**. PP: Probable positives identified by PULSNAR; PN: Probable negatives identified by PULSNAR; CP: coded positives.

training on small, single-institution datasets and reliance on hard-coded NLP rules, which lack generalizability to diverse linguistic patterns and contexts.

## II. DATA COLLECTION AND COHORT DEVELOPMENT

This study presents a comprehensive retrospective cohort analysis utilizing data from the US MarketScan Commercial Claims and Encounters (CCAE) database to estimate the prevalence of OUD employing a PU learning approach. Figure 1 illustrates the methodological steps undertaken in the study, which involve determining the cohort, defining the OUD phenotype, and applying a PU learning technique to estimate the proportion of OUD among undiagnosed individuals within the cohort. The overall OUD prevalence is estimated by combining diagnosed and imputed undiagnosed cases.

### A. Data source

In this study, we used patient claims data from the MarketScan CCAE database (2017-2021), formatted in the OMOP CDMv5 (Observational Medical Outcomes Partnership Common Data Model) [36].

### B. Cohort selection

The cohort for this study was selected based on the following inclusion criteria: (i) individuals must have $\geq 2$ years of observation after January 1, 2017, within the database, and (ii) individuals must have a prescription fill for at least one opioid from the following list: *alfentanil, alphaprodine, buprenorphine, butorphanol, codeine, dextromoramide, dezocine, dihydrocodeine, diphenoxylate, ethylmorphine, fentanyl, heroin, hydrocodone, hydromorphone, levomethadyl, levorphanol, meperidine, meptazinol, methadone, methadyl acetate, morphine, nalbuphine, normethadone, opium, oxycodone, oxymorphone, papaveretum, pentazocine, phenazocine, phenoperidine, pirinitramide, propoxyphene, remifentanil, sufentanil, tapentadol, tilidine, tramadol.*

Despite the transition from the International Classification of Diseases-Ninth Revision-Clinical Modification (ICD-9-CM)

to ICD-10-CM on October 1, 2015, our database still contained many patients with ICD-9-CM codes in 2016 due to a grace period. To maintain consistency and ensure the uniformity of diagnostic coding, only ICD-10-CM diagnosis codes were included in the analysis, and individuals required an observation period starting January 1, 2017 or later to be included in the analysis. Using these inclusion criteria, we selected a cohort of 3,342,044 individuals comprising diagnosed and undiagnosed cases comprising 11,344,113 years of observation.

### C. Phenotyping and covariate selection

Individuals labeled positive (class 1) for the OUD phenotype had at least one of the following ICD-10-CM codes: F11.1*, F11.2*, and F11.93; the rest were considered unlabeled. For the ML model, we identified 94,668 unique covariates, which included age groups (0-10, 10-20, ..., 90-100), sex, and two classes of features: Condition (ICD10CM) and Drug (RxNorm) with their ancestor codes. For positive examples, covariates (conditions and drugs) were selected from the period before the first OUD coding date. In contrast, all conditions and drugs during the observation period were considered for unlabeled examples. Based on the OUD phenotype definition, 45,019 diagnosed (coded) and 3,297,025 unlabeled examples were identified within the study cohort.

To develop the ML model, a compressed sparse matrix of dimensions $3,342,044 \times 94,668$ was generated, representing the study cohort covariates. If a particular condition was documented in the patient's record, the corresponding condition covariate was assigned a value of 1 in the matrix; otherwise 0, indicating the absence of that condition. If a specific drug was present in the patient's record, the corresponding drug covariate was assigned the cumulative drug exposure count during the observation period; otherwise, it was set to 0, reflecting no exposure to that particular drug. The age group covariates were binary, set to 1 for the patient's age range and 0 for the other ranges. The sex covariate was assigned to 1 for male patients and 0 for female patients.

## III. MACHINE LEARNING METHOD: POSITIVE AND UNLABELED (PU) LEARNING

In real-world applications, annotating all records is challenging, expensive, and sometimes even impossible [37], [38]. Often, only positive instances are labeled, and PU learning is particularly useful for such applications, as it handles scenarios where only positive instances are labeled. PU learning is a type of semi-supervised binary classification method where the training data comprises labeled positives and a mixture of unlabeled positives and negatives [38]. Most PU learning algorithms rely on the Selected Completely at Random (SCAR) assumption, which posits that labeled positive examples are randomly selected from the set of all positives, i.e. independent of their attributes [39]. However, this assumption often fails in real-world applications due to factors such as sampling bias, coding bias, data heterogeneity, or temporal variations [38]. For example, patients with severe symptoms are more likely to be diagnosed in medical studies, while those with milder or no symptoms may not be captured. The stigma associated with undesirable behaviors among patients and providers may hinder the full disclosure of information, leading to underdiagnosis. Additionally, biases introduced by the individuals responsible for labeling data can lead to the underrepresentation of certain subgroups.

### A. Why Positive Unlabeled Learning Selected Not at Random (PULSNAR) was used for this study?

Due to the likely violation of the SCAR assumption in OUD undercoding and underdiagnosis, applying SCAR-based PU methods to OUD data can lead to incorrect estimates of OUD prevalence. Consequently, we employed the PULSNAR method [38], which is designed for non-SCAR applications. This method not only estimates the proportion ($\alpha$) of positive examples in the unlabeled set but also provides calibrated predictions, which are essential for identifying at-risk individuals and selecting screening cutoffs in public health efforts.

In scenarios where the SCAR assumption does not hold, the PULSNAR method employs a divide-and-conquer strategy. It divides the coded positive examples into clusters, each predominantly comprising one subtype of positive instances. Subsequently, it applies the SCAR-based method, PULSCAR (Positive Unlabeled Learning Selected Completely At Random), [38] to the positive examples from each cluster and all unlabeled examples to estimate the proportion, $\alpha$, of the positive subtype represented by that cluster. The final $\alpha$ value for all unlabeled observations is obtained by summing the individual $\alpha$ values computed for each cluster.

As demonstrated in [38], PULSNAR produced more accurate $\alpha$ estimates on benchmark datasets compared to other SCAR-based PU methods (e.g., DEDPUL [40], KM1/KM2 [41], TiCE [42]). Additionally, PULSNAR demonstrated superior runtime performance on large datasets versus SCAR-based PU methods. When we tried to apply DEDPUL, KM, and TiCE methods to our large OUD data, they either failed to execute or were too slow to assess confidence intervals with repeated runs, further reinforcing the suitability of PULSNAR.

### B. Alpha estimation, probability calibration, and classification performance using PULSNAR in imbalanced data

Algorithms trained on imbalanced data often exhibit a bias toward the majority class, resulting in poorly calibrated models, overfitting, and poor performance on minority classes. This can also adversely affect model generalizability to new, unseen data [43]. To address this, we avoided applying PULSNAR directly to our highly imbalanced OUD dataset, which has a $|Unlabeled|/|Positive|$ ratio of approximately $k = 73$. Instead, we created 73 balanced datasets by selecting all positive (labeled) examples and a similar number of unlabeled examples, sampled without replacement. PULSNAR was then applied to these 73 balanced datasets to estimate $\alpha_i (1 \leq i \leq 73)$ and calibrate predictions for the unlabeled examples.

The PULSNAR algorithm identified three clusters among the labeled positive examples within each balanced dataset, estimating a separate $\alpha_{ij} (1 \leq j \leq 3)$ for each cluster. The

overall $\alpha_i$ for each balanced dataset was then obtained by summing the $\alpha_{ij}$ values over the clusters.

Using this overall $\alpha_i$, the algorithm generated calibrated predictions for the unlabeled examples within each balanced dataset. We then sorted the unlabeled examples in descending order of the calibrated predictions and selected the top $\alpha_i|U_i|$ examples as probable positives.

To assess classification performance, we trained and tested an XGBoost [44] model using 5-fold cross-validation (CV) on each balanced dataset. The OUD dataset used in this study comprises only labeled positive examples and unlabeled examples, lacking labeled negative examples as ground truth. To address this limitation and ensure a fair comparison between PULSNAR and non-PULSNAR approaches, we employed two distinct modeling strategies. Model 1 was trained and evaluated using 5-fold CV with labeled positives as class 1, and all unlabeled examples as class 0. In contrast, Model 2 was trained and evaluated using 5-fold CV with both labeled positives and $\alpha_i|U_i|$ probable positives as class 1, and the remaining $(1 - \alpha_i)|U_i|$ probable negatives as class 0. To compute the classification performance metrics for both models, we excluded the $\alpha_i|U_i|$ probable positives identified by PULSNAR. This approach ensures that the performance metrics for both models are based on the same data, providing an equitable comparison. All steps to estimate $\alpha$, calibrate predictions, and compute classification performance using the 73 balanced datasets were repeated 40 times to compute 95% confidence intervals (CIs) for the estimates.

The final $\alpha$ for the full dataset was computed by averaging the 73 individual $\alpha_i$ estimates for the balanced datasets. The following derivation elucidates the rationale behind using the mean of the $\alpha$ estimates obtained from each of the 73 balanced datasets to determine the final $\alpha$ for the full dataset.

$$\alpha = \frac{\sum_{i=1}^{k} \alpha_i |U_i|}{|U|} \quad \text{and} \quad |U| = \sum_{i=1}^{k} |U_i|$$

where, $|U|$= number of unlabeled records in full dataset $|U_i|$= number of unlabeled records in balanced dataset $k$= number of balanced datasets

Since $|U1|=|U2|= \ldots. =|U_k|$, $|U|=k|U_k|$

Therefore, $$\alpha = \frac{|U_k| \sum_{i=1}^{k} \alpha_i}{k|U_k|} = \frac{\sum_{i=1}^{k} \alpha_i}{k} \quad (1)$$

### C. Determining important covariates used by the XGBoost model

To identify the covariates that contributed most significantly to the predictive performance of the XGBoost model, we utilized the model-computed gain score for all covariates. The gain score in the XGBoost algorithm measures a feature's relative contribution to the model's predictions. Features with higher gain values are deemed more important, as they lead to substantial reductions in the loss function, thereby enhancing the model's overall performance. For each model trained on the balanced datasets, we selected the important covariates,

TABLE I
**CHARACTERISTICS OF PATIENTS WITH AND WITHOUT CODED OUD.**
These comorbidities are from the list of top important features selected by XGBoost to learn models.

| Patient Characteristics (n=3,342,044) | Coded for OUD (n=45,019) | Uncoded for OUD (n=3,297,025) |
|---|---|---|
| Male | 22,858 (51%) | 1,415,363 (43%) |
| Female | 22,161 (49%) | 1,881,662 (57%) |
| Age, yr | 42 ($\pm$13) | 38 ($\pm$15) |
| **Age, n(%)** | | |
| 0-19 | 1,896 (4.21%) | 459,981 (13.95%) |
| 20-29 | 6,606 (14.67%) | 534,020 (16.20%) |
| 30-39 | 10,590 (23.52%) | 671,288 (20.36%) |
| 40-49 | 11,008 (24.45%) | 688,924 (20.90%) |
| 50-59 | 12,249 (27.21%) | 761,847 (23.11%) |
| >=60 | 2,670 (5.93%) | 180,965 (5.49%) |
| **Comorbidities** | | |
| Chronic pain syndrome + Chronic pain, not elsewhere classified | 18,294 (40.64%) | 444,052 (13.47%) |
| Alcohol related disorders | 4,107 (9.12%) | 89,620 (2.72%) |
| Mental and behavioral disorders | 27,182 (60.38%) | 1,581,777 (47.98%) |
| Other psychoactive substance related disorders | 2,767 (6.15%) | 17,753 (0.54%) |
| Alcohol dependence | 2,926 (6.50%) | 41,913 (1.27%) |
| Bipolar disorder | 2,822 (6.27%) | 65,425 (1.98%) |
| Cannabis related disorders | 2,077 (4.61%) | 44,174 (1.34%) |
| Other stimulant related disorders | 981 (2.18%) | 8,025 (0.24%) |

along with their respective gain scores. Subsequently, the final gain score for each covariate was calculated by taking the mean of its gain scores across all models. Identifying important covariates enabled us to assess which covariates are strongly associated (positively or negatively) with the OUD phenotype, thus providing insights into the underlying risk factors and potential predictors of this condition.

## IV. RESULTS

Applying our inclusion and exclusion criteria, we identified 3,342,044 individuals (1,438,221 males and 1,903,823 females) for the study. Of these, 45,019 (1.35%) were diagnosed with OUD, while the remaining 3,297,025 (98.65%) did not have a coded diagnosis of OUD. Table I provides a summary of patient characteristics. The average age of patients with coded OUD was 42 years (standard deviation: 13), while the average age of those without coded OUD was 38 years (standard deviation: 15).

### A. Alpha estimates by PULSNAR

In a cohort of 3,342,044 individuals, only 45,019 cases (1.35%) were diagnosed with OUD. The PULSNAR method estimated an additional 124,723 cases of OUD among unlabeled individuals (3.78% of unlabeled individuals, imputed OUD, 95% CI: [3.76%, 3.80%]), with a cumulative prevalence over 2-5 years and a mean observation period of 3.39 years. Consequently, the overall cumulative prevalence of OUD, combining both diagnosed and imputed undiagnosed cases, was 5.08% across all age groups and sexes, with 73.5% of the cases being imputed. Figure 2 displays a histogram of the mean $\alpha$ estimates obtained using the PULSNAR method for each iteration based on 73 balanced datasets, along with the corresponding 95% confidence interval (CI).

### B. Classification performance with and without PULSNAR

In all 40 iterations, XGBoost classification performance improved across each of the 73 balanced datasets when probable positives (imputed cases) identified using the PULSNAR method were included as positive instances for training

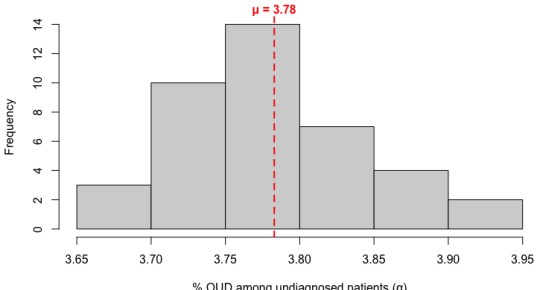

Fig. 2. **Distribution of $\alpha$ estimates by PULSNAR method**. Each iteration had 73 $\alpha$ estimates, each corresponding to one of the 73 balanced datasets. The mean $\alpha$ estimate across all iterations was 0.0378 (3.78% of the 3,297,025 unlabeled are positive), with 95% CI: [0.0376, 0.0380]. Red line: mean $\alpha$ value

and testing the model. Table II presents the classification performance of the XGBoost model with probable positives kept with the unlabeled vs. placed among the positives. The reported classification metric values represent the mean and 95% CI based on 2,920 models (73 balanced datasets × 40 iterations).

### C. Characterization of OUD pattern by state

The coded fraction of OUD, defined as the proportion of explicitly coded cases relative to the total of diagnosed and imputed undiagnosed cases, across US states is shown in Figure 3. A higher coded fraction in a state indicates a greater likelihood of individuals with OUD receiving diagnoses within that state. Conversely, a lower coded fraction suggests a higher proportion of undiagnosed cases. The coding of OUD among US states varied considerably, ranging from 14.49% in Nebraska to 49.31% in Delaware.

The diagnosed (coded) fraction, defined as the proportion of individuals with a documented OUD diagnosis among opioid users, ranged from 0.58% in Nebraska to 3.29% in Delaware (Figure 4). Furthermore, the combined prevalence of OUD, encompassing both coded and imputed cases, ranged from 2.79% in Washington D.C. to 10.60% in Idaho (Figure 4).

### D. Characterization of OUD pattern by sex

In a cohort of 3,342,044 individuals, 43% were male, and 57% were female. The prevalence of coded OUD was 1.59% among males and 1.16% among females. However, when including both diagnosed cases and imputed undiagnosed cases identified through the PULSNAR method, the overall prevalence of OUD was estimated to be 5.48% for males and 4.62% for females. Additionally, the coded fraction revealed a statistically significant difference in the likelihood of receiving an OUD diagnosis between sexes. Males were more likely to have a coded OUD diagnosis than females, with respective percentages of 29.02% and 25.22% ($p = 4.79e^{-68}$). Table III shows the distribution of individuals and OUD metrics by sex.

### E. Characterization of OUD pattern by age

Figure 5 shows the distribution of OUD among opioid users across different age groups in the US. The diagnosed

TABLE II

**CLASSIFICATION PERFORMANCE OF XGBOOST MODELS**. Probable positives identified by the PULSNAR method (imputed cases) contributed to improved XGBoost classification. AUC-ROC: Area under the Receiver Operating Characteristic Curve; MCC: Matthews correlation coefficient; APS: Average precision score; BS: Brier score loss, PPV: Positive predictive value. Model 1: positives as class 1 and unlabeled as class 0, Model 2: labeled + probable positives as class 1, probable negatives as class 0

| Performance metric | Model 1 | Model 2 |
|---|---|---|
| AUC-ROC | 0.9692 (0.9691, 0.9692) | 0.9706 (0.9706, 0.9707) |
| Accuracy | 0.9076 (0.9075, 0.9076) | 0.9105 (0.9104, 0.9105) |
| MCC | 0.8168 (0.8166, 0.8170) | 0.8215 (0.8213, 0.8217) |
| APS | 0.9745 (0.9744, 0.9745) | 0.9757 (0.9757, 0.9758) |
| BS | 0.0676 (0.0675, 0.0677) | 0.0649 (0.0648, 0.0650) |
| F1 | 0.9064 (0.9063, 0.9065) | 0.9107 (0.9106, 0.9108) |
| Sensitivity | 0.8790 (0.8790, 0.8791) | 0.8939 (0.8938, 0.8941) |
| Specificity | 0.9371 (0.9370, 0.9373) | 0.9276 (0.9275, 0.9277) |
| PPV | 0.9355 (0.9354, 0.9355) | 0.9276(0.9275, 0.9277) |

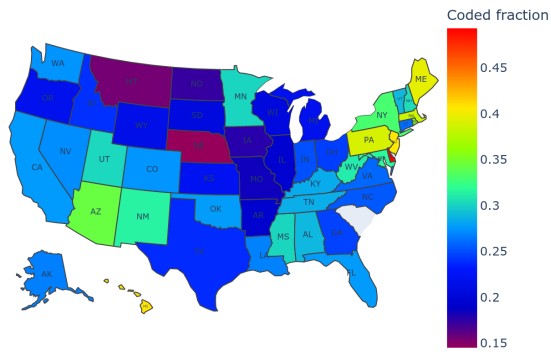

Fig. 3. **Fraction of coded OUD by US states**. Due to MarketScan license restrictions, data for South Carolina were excluded from the figure. Coded fraction=coded/(coded+imputed). PULSNAR imputation suggests the fraction of OUD coded ranges from 14.49-49.31%.

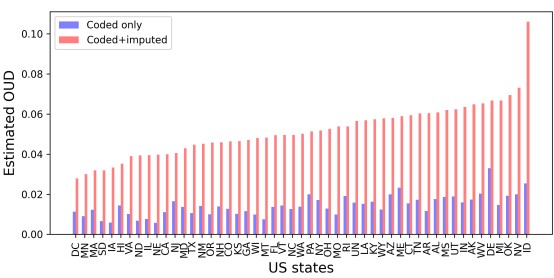

Fig. 4. **Estimated OUD among opioid users by US states**. Coded plus imputed OUD fraction among those who had at least one opioid prescription filled ranged from 2.79-10.60% across US states. South Carolina was excluded, as described in Figure 3. UN=Unknown.

TABLE III

**DISTRIBUTION OF INDIVIDUALS AND OUD PATTERN BY SEX.** Coded fraction = coded/(coded+imputed) and coded+imputed represents the percentage of coded+imputed by sex.

| Sex | Total count | Coded | Imputed | Coded+ imputed | Coded fraction |
|---|---|---|---|---|---|
| Male | 1,438,221 | 22,858 | 55,918 | 5.48% | 29.02% |
| Female | 1,903,823 | 22,161 | 65,723 | 4.62% | 25.22% |

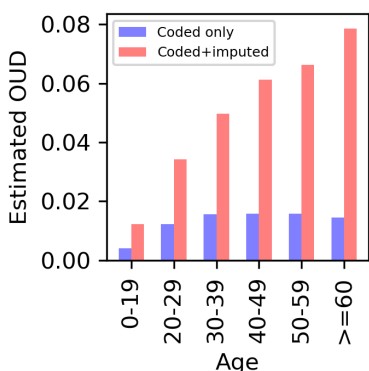

Fig. 5. **Estimated OUD among opioid users by age**. Coded plus imputed OUD fraction among those who had at least one opioid prescription filled ranged from 1.22-7.86% across age groups.

(coded) fraction ranged from 0.41% to 1.58%. The combined prevalence of OUD, encompassing both coded and imputed cases, ranged from 1.22% to 7.86% over the age groups, with older ages associated with higher prevalence.

### F. Important covariates used by the XGBoost classifier

Out of the 94,668 covariates available in our dataset, only 10,190 (10.76%) were utilized by the XGBoost algorithm to learn from the data, comprising coded positives and probable positives and negatives identified by the PULSNAR method. The top 15 covariates, as determined by their gain scores, are presented in Table IV. These covariates are based on 2,920 XGBoost models (73 balanced datasets × 40 iterations), with their gain score averaged over all models. The table also presents the fraction of patients who had coded OUD and the fraction of patients who had imputed OUD, with and without each covariate. Patients with covariates such as buprenorphine, chronic pain syndrome, naloxone, and other psychoactive substance-related disorders were more likely to be predicted as having OUD by the model. Conversely, the model is more likely to predict a lower probability of OUD for individuals with certain covariates, such as the outcome of delivery, COVID-19 diagnosis, or COVID-19 vaccination.

## V. DISCUSSION

Estimating the true prevalence of OUD is crucial for guiding public health policies, designing effective interventions and screening programs, allocating resources, identifying health disparities, and mitigating the harms of this condition. However, existing methods, including general population surveys and patient health data analyses, often underestimate the true prevalence due to underreporting, underdiagnosis, and limited healthcare access [13]–[18], [45].

This study highlights the utility of PULSNAR for imputing uncoded or undiagnosed OUD cases, even when the SCAR assumption does not hold. The coded cumulative prevalence of OUD in our cohort was low, with only 1 in 73 individuals diagnosed. After applying the PULSNAR method, we estimated that approximately 1 in 20 individuals have OUD, resulting in

TABLE IV
**TOP 15 COVARIATES USED BY XGBOOST TO LEARN THE MODEL.** The gain score reflects the magnitude of the covariate contribution to the model's prediction. [1]percentage of coded OUD patients with the corresponding covariate set to zero. [2]percentage of coded OUD patients with the corresponding non-zero covariate. [3]percentage of patients with imputed OUD with the corresponding covariate set to zero. [4]percentage of patients with imputed OUD with the corresponding non-zero covariate.

| Covariate | % coded OUD (Covariate = 0)[1] | % coded OUD (Covariate > 0)[2] | % imputed OUD (Covariate = 0)[3] | % imputed OUD (Covariate > 0)[4] | Gain Score |
|---|---|---|---|---|---|
| Symptoms, signs and abnormal clinical and laboratory findings, not elsewhere classified | 3.64 | 1.04 | 3.29 | 3.69 | 383.03 |
| Acetaminophen | 2.26 | 0.99 | 2.59 | 4.05 | 356.65 |
| SARS-CoV-2 (COVID-19) vaccine, mRNA spike protein | 2.08 | 0.12 | 5.03 | 1.31 | 312.74 |
| Buprenorphine | 1.11 | 38.07 | 3.43 | 37.53 | 281.95 |
| Chronic pain syndrome | 1.14 | 9.36 | 2.85 | 34.61 | 253.62 |
| Diseases of the respiratory system | 2.25 | 0.92 | 3.71 | 3.61 | 249.91 |
| Alcohol related disorders | 1.26 | 4.38 | 3.39 | 12.28 | 239.47 |
| Outcome of delivery | 1.42 | 0.24 | 3.82 | 0.95 | 200.75 |
| Fentanyl | 1.55 | 0.81 | 3.59 | 3.77 | 191.42 |
| Mental and behavioural disorders | 1.03 | 1.69 | 1.78 | 5.64 | 168.71 |
| Other psychoactive substance related disorders | 1.27 | 13.48 | 3.51 | 24.37 | 163.35 |
| Diseases of the skin and subcutaneous tissue | 1.97 | 0.76 | 3.97 | 3.33 | 152.13 |
| naloxone | 1.12 | 19.5 | 3.32 | 29.56 | 150.33 |
| codeine | 1.53 | 0.81 | 3.56 | 3.87 | 144.8 |
| Emergency use of U07.1, COVID-19, virus identified | 1.49 | 0.23 | 3.85 | 2.01 | 131.95 |

an overall estimated cumulative prevalence of 5.08% across all age groups and sexes. While we could not verify the imputed cases due to the absence of clinical notes in our claims data, our estimate aligns well with the 2-5% range reported in other studies [45]–[47]. Furthermore, according to [46], the OUD prevalence across US states ranges from 0.6-9.7%, with the upper limit of this range corroborating our findings (Figure 4). To further validate the reliability of PULSNAR, in a separate study, we applied the method to identify uncoded self-harm cases in the EHR data of US veterans. The identified cases were subsequently validated through a chart review of their clinical notes.

Existing literature has indicated a higher prevalence of OUD among males compared to females, potentially due to the elevated rates of opioid misuse observed in the male population [48]. This pattern is reflected in our findings, with males having a higher prevalence and a greater likelihood of being coded than their female counterparts. Specifically, the coded fraction for OUD was significantly higher among males than females (29.04% vs. 25.22%). Furthermore, the difference in coded fractions between males and females suggests potential biases or disparities in diagnostic practices, healthcare access, and/or help-seeking behaviors.

The observed disparities in OUD estimates (coded+imputed), ranging from 2.79% in Washington D.C. to 10.60% in Idaho, may reflect differences in coding practices and public health policies within US states. Nebraska had the lowest coded fraction at 14.49%, suggesting a higher prevalence of undiagnosed cases. In comparison, Delaware had the highest coded fraction at 49.31%, suggesting more rigor in detection or coding of OUD. A potential uncorrected source of bias in PULSNAR-imputed OUD across US states could be variability in naloxone availability for harm reduction. Some states may have more persons obtaining naloxone prescriptions for family members and acquaintances with dependence, and the model might ascribe OUD to them

with a higher probability. The high additional imputed levels of OUD within Idaho may be an artifact of the small sample size for that state (2,045 people with opioid exposure).

According to previous studies [47], [49], [50], older populations are less likely to receive a clinical diagnosis or coding for OUD compared to younger adults despite having problematic opioid use patterns. This observation aligns with the findings of our study, which demonstrated increasingly higher imputed levels of OUD with increasing age.

The top important features identified by the XGBoost models provide valuable insights into the multifaceted nature of OUD and its associated risk factors. Notably, medications such as buprenorphine and naloxone, commonly used in treating and managing OUD, emerged as top features. Additionally, the model highlighted comorbidities, including chronic pain syndrome, alcohol-related disorders, and other psychoactive substance-related disorders in the context of OUD. While the COVID-19 vaccine and diagnosis emerged as top features in our model, these features correspond to a lower predicted probability of OUD for individuals (see Table IV). This inverse relationship may be due to lack of access to addiction care during the pandemic, as evidenced by the 17.3:1 ratio of coded OUD unvaccinated to vaccinated vs. the 3.8:1 ratio of imputed OUD unvaccinated to vaccinated. That is, OUD may have gone more undetected during the pandemic.

Acetaminophen is frequently prescribed for pain relief and fever reduction, sometimes in combination with opioid analgesics for more severe pain management [51], perhaps explaining its emergence as a top feature. The lower OUD rates among those prescribed acetaminophen may be associated with its use for mild-to-moderate pain, which does not require opioids.

**Implications.** Our findings demonstrate the potential of PU learning to estimate the prevalence of undiagnosed OUD, which is key for effective public health initiatives and screening efforts. By addressing undercoding and underdiagnosis, our approach can mitigate the harms of OUD and improve resource allocation. Policymakers can use accurate prevalence data to create policies and allocate resources to areas with higher rates of undiagnosed OUD. Healthcare providers can enhance screening and diagnostic procedures, and insurance companies may expand coverage based on these insights. The study also highlights the promise of machine learning tools in improving disease tracking and management.

**Limitations and Future Work.** While our study demonstrates the effectiveness of PU learning, it is limited by the reliance on claims data, which may not capture all relevant information and may not represent the entire state's population. Future research should explore integrating other data sources, such as electronic health records and multi-modal data, to enhance prediction accuracy. Since most PU methods are based on the SCAR assumption, and their publicly available implementations are unsuitable for large datasets, we could not compare the $\alpha$ estimate obtained from PULSNAR with other methods. Additionally, the study population consists of US commercially insured patients, limiting the generalizability of

findings. Data from 2017 to 2021 cannot account for changes in opioid use patterns, diagnostic practices, or healthcare policies after this period. Diagnostic coding practices vary across providers and states and could introduce bias. Moreover, the health claims data may not fully capture comorbid conditions. Future research should address these limitations. Clinical validation of imputed OUD cases is crucial for validating PULSNAR. Including uninsured and publicly insured populations would improve generalizability. Incorporating socioeconomic status and healthcare access into the analysis would clarify factors influencing OUD diagnosis and prevalence.

**Conclusion.** This study demonstrates the utility of PU learning in addressing underdiagnosis and undercoding of OUD in health claims data. Our findings highlight demographic and geographic disparities in OUD diagnosis, underscoring the need for better diagnostic practices and resource allocation. PULSNAR holds promise for enhancing public health strategies and targeted interventions for the opioid epidemic, as well as an array of other healthcare challenges.

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
