# OpenReview forum: "Detecting Opioid Use Disorder in Health Claims Data with Positive Unlabeled Learning"
_IEEE.org/EMBS/BHI/2024/Conference — IEEE BHI'24_

### Official Review · Reviewer_rsct · 2024-08-10
**The paper is a good contribution on the prediction of the opioid use disorder cases using non-SCAR based machine learning method.**

**Overall Rating:** 7
**Confidence:** 3

**Other Quality Metrics:**

(a) Clarity of writing: excellent
(b) Clinical Significance: excellent
(c) Methodological Novelty: good
(d) Experiments and Results: excellent

**Questions For The Authors:**

In table 2 where the performance of the 2 models are evaluated, the ground truth used for either of the 2 models is a bit confusing to me. Since the labels used for the training of the 2 models are different, are the ground truths also different for performance evaluation? And if so, is the comparison between the performance reasonable?

**Strengths:**

1. The reasoning of the choice of the PULSNAR method is persuasive.
2. The prediction of OUD prevalence resulting from the proposed method is compared to many related researches and well discussed.
3. Discussion on the covariates is valuable for further studies on the subject.

**Summary Of The Paper:**

The paper applies a non-SCAR based machine learning method to the prediction of the likely OUD cases in the US. The results show that a significant proportion of uncoded cases are likely to have OUD problems and demonstrates consistency with related researches.

**Weaknesses:**

Some comparison to existing methods would be nice, but this issue has been acknowledged in the discussion section.

---

### Official Review · Reviewer_4wYj · 2024-08-11
**Detecting Opioid Use Disorder in Health Claims Data with Positive Unlabeled Learning**

**Overall Rating:** 7
**Confidence:** 1

**Other Quality Metrics:**

Clarity of Writing : Good
Clinical Significance : Somewhat significant
Methodological Novelty : Adequately Novel
Experiments and Results : Good

**Questions For The Authors:**

None

**Strengths:**

Well written paper adressing an important issue in bias introduced  by unlabeled data assumed as negative or positive labeled  during machine learning.   The literature review is detailed and results are promising.

**Summary Of The Paper:**

This study estimates Opioid use using patient claims data from the Mar- ketScan CCAE database.  They highlight  that patients without a diagnosis labeled as positive or negative  cases may introduce bian  into models effecting models predictive power.  They suggest Positive Unlabeled Learning Selected Not Randiom. (PULSNAR) to estimate the probability of a patient having OUD during a time window.

**Weaknesses:**

The claims data being used introduces selection bias as pointed out by the authors.   The data being only US data also limits the generalizability of the study.

---

### Official Review · Reviewer_uprj · 2024-08-14
**Review 225**

**Overall Rating:** 7
**Confidence:** 3

**Other Quality Metrics:**

Clarity of Writing: Good
Clinical Significance: Good
Methodological Novelty: Fair
Experiments and Results: Fair

**Questions For The Authors:**

Can you provide more details on how you addressed potential biases in coding practices across different states?

**Strengths:**

Well-structured writing, providing important findings regarding the underdiagnosis of OUD and highlights potential disparities in diagnosis rates across different states and demographic groups.

**Summary Of The Paper:**

The authors present a method for detecting OUD in health claims data using a Positive Unlabeled Learning approach PULSNAR technique. The study aims to estimate the prevalence of OUD in a large cohort of commercially insured U.S. patients. The paper addresses the challenges of underdiagnosis and undercoding of OUD in health records. The study indicates that a significant portion of OUD cases are not diagnosed or coded. The study also explores demographic and geographic variations in OUD prevalence and identifies key covariates associated with the disorder using an XGBoost classifier.

**Weaknesses:**

1. The imputed OUD cases identified by the PULSNAR method lack clinical validation.
2. While the paper highlights the advantages of PULSNAR, it does not provide a detailed comparison with other PU learning methods in terms of performance.

---

### Decision · Program_Chairs · 2024-09-23

Accept